# Drug Delivery and Therapy Strategies for Osteoporosis Intervention

**DOI:** 10.3390/molecules28186652

**Published:** 2023-09-16

**Authors:** Mingyang Ma, Huiling Zeng, Pei Yang, Jiabing Xu, Xingwang Zhang, Wei He

**Affiliations:** 1School of Pharmacy, China Pharmaceutical University, Nanjing 211198, China; 15084209372@163.com (M.M.); zenghl923@163.com (H.Z.); 2School of Science, China Pharmaceutical University, Nanjing 211198, China; pharmyp@163.com; 3Taizhou Institute for Drug Control, Taizhou 225316, China; xujiabing2@sina.com; 4Department of Pharmaceutics, School of Pharmacy, Jinan University, Guangzhou 510632, China; 5Shanghai Skin Disease Hospital, Tongji University School of Medicine, Shanghai 200443, China

**Keywords:** osteoporosis, drug delivery, targeted drug delivery, bone tissue engineering, microneedles

## Abstract

With the advent of the aging society, osteoporosis (OP) risk increases yearly. Currently, the clinical usage of anti-OP drugs is challenged by recurrent side effects and poor patient compliance, regardless of oral, intravenous, or subcutaneous administration. Properly using a drug delivery system or formulation strategy can achieve targeted drug delivery to the bone, diminish side effects, improve bioavailability, and prolong the in vivo residence time, thus effectively curing osteoporosis. This review expounds on the pathogenesis of OP and the clinical medicaments used for OP intervention, proposes the design approach for anti-OP drug delivery, emphatically discusses emerging novel anti-OP drug delivery systems, and enumerates anti-OP preparations under clinical investigation. Our findings may contribute to engineering anti-OP drug delivery and OP-targeting therapy.

## 1. Introduction

Osteoporosis (OP) is a progressive systemic skeletal disorder characterized by decreased bone mass, deterioration of bone microstructure, decreased bone strength, and increased fracture risk [1,2]. Patients with mild osteoporosis have no apparent symptoms, and as the disease progresses, they may experience waist and back pain or even generalized bone pain [3]. In addition, patients are more likely to fracture when they fall, and severe osteoporosis can also lead to bone deformation, such as a hunched back, which is one of the leading causes of disability and death in elderly patients, seriously affecting the physical and mental health of middle-aged and older people [4]. Because menopausal estrogen deficiency leads to bone resorption greater than bone formation, older women are more likely to develop OP than older men. Globally, one-third of women and one-fifth of men over 50 are susceptible to osteoporotic fractures, with the most common osteoporosis-related fractures occurring in the spine, hip, and wrist [5]. A recent study by Cui et al. suggests that by 2040, the proportion of people over 50 in China will increase by about 100%, while the number of fragility fractures in China will increase by 135% per year [6]. With the aging of society, the increasing incidence of osteoporosis, plus serious consequences such as fractures, the World Health Organization has identified OP as a significant public health problem [7]. Effective management and treatment of OP are becoming an urgent medical task requiring the development of novel medicaments.

In recent years, the ideology of drug delivery has gained ground, and is well-placed to overcome various physiological barriers to effectively intervening in intractable diseases. Various innovative anti-OP drug delivery systems have enabled successful treatment of the disease of OP. In this review, we discuss the pathogenesis of OP and therapeutic medications thereof. In addition, several emerging therapeutic strategies are presented, including ligand modification of nanoparticles for targeted delivery of anti-OP drugs, development of supersaturated drug delivery systems, i.e., liposomes and nanoemulsions for solubilization of poorly water-soluble drugs, adoption of non-invasive delivery methods such as oral and transdermal delivery to improve patient compliance, and coupling of drugs with scaffolds for repair of bone defects and local drug delivery. In particular, we highlight the applications of novel drug delivery systems in the fight against OP.

## 2. The Pathophysiology of OP and Clinical Medicaments

Bone homeostasis refers to the balance maintained by the body between bone resorption and bone formation. Bone homeostasis imbalance tends to emerge as OP. Bone resorption and bone formation allow the body’s bone tissue to be constantly renewed. Osteoblasts are the main functional bone formation cells responsible for synthesizing, secreting, and mineralizing the bone matrix [8]. Osteoclasts are derived from hematopoietic stem cells in the bone marrow. Multinucleated cells secrete acidic substances and various proteases (such as cathepsin K) to digest the bone matrix and form bone resorption fosses.

Various molecules coordinate osteoblast and osteoclast functions to maintain bone homeostasis, in which OPG/RANK/RANKL is a system that regulates the differentiation and maturation of osteoclasts [9]. Osteoblasts are derived from bone marrow mesenchymal stem cells (BMSCs), and Runx2 is a key transcription factor for BMSCs and pre-osteoblast-to-osteoblast differentiation. Runx2 is regulated by various signals and pathways, such as bone morphogenetic proteins (BMPs) and the Wnt/β-catenin signaling pathway. BMP2 and BMP4 phosphorylate SMAD1/5/8 and activate Runx2 to stimulate osteoblast differentiation [10]. Wnt proteins (especially Wnt3a and Wnt10b) and Frizzled family proteins bind to low-density lipoprotein receptor-related protein 5/6 to increase the level of Runx2 in the body, thereby promoting the differentiation of osteoblasts [11]. An osteosclerotic protein secreted by osteocytes binds to LRP-5/6 and inhibits the Wnt/β-catenin signaling pathway, decreasing Runx2 expression [12]. Osteoclast precursors express receptor activators for nuclear factor κ-B (RANK). RANK binds to the receptor activator of nuclear factor κ-B ligand (RANKL) expressed on osteoblasts and osteocytes to promote osteoclasts differentiation. Osteoblasts secrete osteoprotegerin (OPG), an induction receptor for RANKL, and OPG can compete with RANK to bind to RANKL and thus inhibit osteoclasts differentiation [13]. Although BMSCs have the potential to differentiate into osteoblasts, they can also differentiate into adipocytes. In some cases, such as menopause, aging, and long-term glucocorticoid use, osteoclasts will be overactivated, osteoblasts’ function will be inhibited, osteogenic differentiation of BMSCs will be weakened, adipogenic differentiation will be enhanced, and the formation of bone blood vessels will be blocked, eventually leading to OP [14].

Peak bone mass in women is lower than in men, which may be why women are more susceptible to OP than men [15]. In addition, various physiological periods in women can also lead to bone loss, such as pregnancy, lactation, menopause, etc. [16]. The role of hormones in OP development cannot be underestimated. Decreased levels of sex steroids, particularly estradiol, can mediate age-related bone loss in men and women [17,18]. The sex steroid hormone that disturbs bone metabolism in women is estrogen. After binding to estrogen receptors, estrogen induces apoptosis of osteoclasts through the Fas/FasL system and exerts anti-OP effects [19]. The hormones that affect male bone metabolism include testosterone, free testosterone, and estradiol. Free testosterone acts on androgen receptors, promotes bone formation, inhibits bone resorption, and alters bone metabolism. Testosterone affects bone by being converted into estrogen by aromatase. Hypotestosteronism promotes RANKL secretion and stimulates osteoclast proliferation [20].

Based on the pathogenesis of OP, its therapeutic agents can be divided into two classes [21,22,23]. One class is bone resorption inhibitors such as calcitonin, estrogen, and selective estrogen receptor modulators (SERMs), bisphosphonates, and RANKL inhibitors. The other category is bone formation promoters, such as parathyroid hormone and its analogs and sclerostin inhibitors. Statins are drugs used to lower serum cholesterol. Male patients with OP and comorbid hypogonadism can be treated with testosterone therapy overlaid with medication [24]. Recent studies have disclosed that simvastatin can increase BMP2 expression in osteocytes, while lovastatin, simvastatin, mevastatin, and fluvastatin significantly promote rodent bone formation [25]. In this regard, it can be seen that statins can be used as potential drugs for OP therapy, but they have not been used clinically to date. Recently, some researchers have found that miRNA-based gene regulation can treat OP. Increasing evidence has shown that miRNAs regulate bone metabolism and modulate the proliferation and differentiation of BMSCs, osteoblasts, and osteoclasts, which play an important role in maintaining bone homeostasis [26]. It has been reported that the occurrence and development of elderly OP and postmenopausal OP are related to high expression of miR214 [27]. Expression of miR214 inhibits bone formation and promotes osteoclast production [28]. In addition, some traditional Chinese medicines, such as icariin, also have anti-osteoporotic effects [29]. Figure 1 describes the mechanism of action of different anti-OP drugs, and Table 1 generalizes the anti-OP drugs used in the clinical stage.

Even though many drugs have been used to treat OP, some side effects and non-compliance associated with dosage form generally compromise the use of such drugs. Oral bisphosphonates are severely irritating to the esophagus, require a large amount of water to swallow, and patients must then avoid lying down for half an hour [30]. In addition, long-term use of bisphosphonates may result in side effects, such as jaw osteonecrosis and atypical femoral fracture [31]. Deposit of estrogen in non-skeletal sites increases the risk of breast cancer and induces adverse cardiovascular effects in women [32]. SERMs, such as raloxifene, have good safety. Studies of postmenopausal women at high risk of cardiovascular disease indicated that raloxifene did not elevate the risk of coronary artery disease and stroke. Furthermore, raloxifene acts as an estrogen antagonist in the mammary glands and uterus and, thus, does not stimulate the mammary glands and uterus [33]. However, certain studies reported that the drug mildly increases the risk of venous embolism, limiting its use in patients with a history of venous embolism and thrombotic tendencies [34]. Moreover, raloxifene is not indicated for men with OP. Additionally, testosterone has several side effects, such as cardiovascular disease, erythrocytosis, water retention, and increased plasma fibrinolytic enzymes [35]. Calcitonin injection is not convenient for patients requiring long-term medication, and the patient’s compliance is significantly reduced due to pain or needle phobia. Calcitonin nasal sprays also have side effects, such as epistaxis, rhinitis, and nasal mucosal ulcers [36]. The oral bioavailability of strontium ranelate is insufficient. Long-term use increases the risk of cardiovascular diseases and adverse reactions such as nausea, diarrhea, venous thrombosis, headache, and even dermatitis may occur [37]. These factors have contributed to a shift to developing more efficient and safer drug delivery systems. Figure 2 illustrates several emerging anti-OP drug delivery systems.

## 3. Engineering of Anti-OP Drugs Delivery

### 3.1. Targeted Delivery of Anti-OP Drugs

For estrogen-based drugs, the rational design of targeted delivery vehicles can reduce or avoid drug distribution toward non-target tissues and premature metabolism and diminish the side effects caused by high systemic exposure of drugs [38]. Targeted ligands with a specific affinity for bone-associated cells or the bone matrix hold promise for enhancive drug accumulation in the bone [39]. Hydroxyapatite (HA), the main component of bone matrix with a molecular weight of 1000 Da, has good biocompatibility and bone affinity, and most drugs achieve bone-targeted delivery through targeting HA [40,41]. Figure 3 shows three commonly used ligands for targeting HA. Bisphosphonates are chemically stable derivatives of natural pyrophosphate, structurally containing two closely linked phosphate groups (P-C-P). The hydroxyl deprotonation in these two phosphonate groups can chelate with Ca^2+^ on HA to form a bidentate structure and firmly bind HA, so bisphosphonates have promising bone-targeting potential [42]. For example, Guo et al. [43] prepared poly (lactic-co-glycolic acid) (PLGA) nanoparticles for the synchronous delivery of estradiol and iron oxides to the bone. The surface of the particles was modified with alendronate, a ligand able to target the bone matrix. The delivery system can control drug release in response to an external magnetic field in the presence of iron oxide. The experimental results showed that ligand-anchored nanoparticles could be availably assimilated by Raw 264.7 cells and effectively enriched estradiol in the bone tissue. Notably, the use of bisphosphonates as ligands is potentially risky because bisphosphonates inhibit bone resorption, so prolonged low bone conversion may lead to atypical femur fractures [44]. The enol, β-diketone, and formamide groups in the molecular structure of tetracycline can be chelated with Ca^2+^ on HA; thus, tetracycline can also be used as a targeting ligand in bone-targeting drug delivery [45]. Tetracycline is relatively nontoxic compared to bisphosphonates, but the complex structure of tetracycline limits its application due to poor stability in chemical coupling [46]. Oligopeptides containing acidic amino acids (e.g., aspartic acid (Asp) and serine (Ser)) can bind tightly to Ca^2+^ on HA and serve as bone-targeting moieties [47]. Among them, oligopeptides such as (Asp-Ser-Ser)_6_, SDSSD (Ser-Asp-Ser-Ser-Asp), and Asp_8_ are the most commonly used species [48]. (Asp-Ser-Ser)_6_ combines strongly with low-crystalline HA and is thus highly selective for the bone-forming surface [49]. The oligopeptide SDSSD specifically binds to osteoblasts via periosteum proteins [48]. Asp_8_ can target highly crystalline HA in the bone resorption region [50]. Additionally, oligopeptides have no long-term adverse effects and are safe as ligands with no synthetic or unnatural chemicals in the delivery system [51].

In addition, appropriate aptamers allow specific-cell targeting and improved drug delivery. Some single-stranded DNA or RNA sequences can target the bone, e.g., CH6 aptamer, which was proven to target human and murine osteoblasts [52]. Furthermore, BMSCs can be targeted by screening suitable aptamers to deliver some drugs that can promote the differentiation of BMSCs to osteoblasts. Zhou et al. combined BMSCs aptamers with nanoparticles loaded with antagomir-483-5p to promote the osteogenic differentiation of BMSCs, providing a new idea for the clinical treatment and prevention of OP [53,54]. Although the emergence of bone matrix and osteoblast-based ligands has dramatically increased the targeting of OP drug delivery, these systems still suffer from off-target side effects. In addition, targeted delivery systems may have a negative impact on immunocompromised patients, requiring a determination of the risk of drug administration to patients [55].

### 3.2. Developing Supersaturated Drug Delivery Systems

Supersaturated drug delivery systems (SDDS) incorporate the drug in a high-energy or fast-dissolving form so that the concentration of the drug exceeds the solubility in the saturated state, e.g., lipid-based delivery systems [56,57], nanoparticle delivery systems [58,59], etc. SDDS are widely used to improve the solubility of poorly water-soluble drugs and have the advantage of improving oral bioavailability and enhancing skin permeability [60]. Formulating poorly water-soluble drugs such as raloxifene, icariin, and tocotrienol into liposomes or micro/nanoemulsions can increase their exposure levels in application sites, hence the bioavailability [61]. For example, Zakir et al. [62] developed a formulation of nanoemulsion-based gels for transdermal delivery of raloxifene. Pharmacokinetic results showed that a 26-fold increase in raloxifene bioavailability was achieved by enabling local supersaturation compared to the marketed oral tablets. Ansari et al. [63] fabricated spanlastic vesicles (a modified version of niosomes) for percutaneous delivery of raloxifene. The results of ex vivo skin permeability studies showed that the elastic vesicles loading raloxifene penetrated deeper into the skin layer than raloxifene suspensions (depth of about 54.9 μm), and raloxifene suspensions were limited to a depth of 8 μm. The above studies show that the supersaturation strategy using nanoparticles or nanovesicles can improve the local bioavailability of poorly soluble anti-OP drugs. Despite the attractiveness of SDDS (e.g., liposomes and nanoemulsions) in improving the solubility of poorly water-soluble drugs, there are still significant challenges in their formulation and manufacturing. For example, difficulty in selecting excipients with both emulsifying properties and high solubility, risk of re-precipitation of formulations due to interaction of lipid excipients with the environment, side effects such as gastrointestinal irritation due to extensive use of surfactants, lack of compressibility and biocompatibility of lipid-based emulsion delivery systems, and difficulty in converting them to final dosage form, etc. [64].

### 3.3. The Use of Non-Invasive Drug Delivery Systems

Patients will feel distressed during long-term medication with peptide drugs requiring an intravenous administration. The design of non-invasive drug delivery systems allows bioactive molecules to be delivered into the body through sublingual, oral, transdermal, and other routes, which can improve patients’ quality of life. Oral drug administration has a higher safety profile, improves patient compliance, and is the preferred route of administration. However, the harsh environment of the gastrointestinal tract and mucus and epithelial barriers lead to challenges in oral drug absorption [65,66]. Particulate-based carriers such as polymeric nanoparticles and liposomes can safeguard peptide drugs from being partially absorbed via the oral route. Based on this assumption, Liu et al. [67] developed chitosan-modified calcitonin and puerarin dual-loaded oral nanoparticles. It was shown that puerarin exhibited particular enzyme-inhibiting ability and the oral absolute bioavailability of calcitonin in the double-loaded nanoparticles was up to 12.52 ± 1.83%, approximately 1.74 times that of pueraria-free nanoparticles.

Besides the oral route, the transdermal way is also an alternative for designing non-invasive delivery systems. Transdermal drug delivery refers to the systemic or local delivery of drugs through the skin at a controlled rate, which is a good alternative to oral and parenteral drug delivery because it is less painful, easy to administer, and easy to stop, and avoids the destruction and first-pass metabolism of the drug in the gastrointestinal tract environment [68]. Despite the great advantages of transdermal drug delivery, there are only a few marketed formulations, which is due to the lower permeability of the stratum corneum [69]. Strategies to overcome skin permeability include microneedles, iontophoresis, electroporation, chemical penetrants, nano-delivery systems, etc. [70]. Microneedles are the most widely used form of drug delivery by the transdermal route, with high patient compliance and no specialized knowledge required [71]. Li et al. [72] prepared a microneedle system consisting of silk fibroin needle tips and hyaluronic acid matrix for transdermal delivery of salmon calcitonin. The developed microneedles showed significant trabecular repair ability, and the therapeutic effect on OVX mice was better than that of traditional injections. Therefore, the microneedles system is acknowledged as promising in long-term transdermal salmon calcitonin delivery for the treatment of OP, which is expected to replace subcutaneous injection with a traditional needle. In addition, for bisphosphonates, engineering a suitable carrier can reduce gastrointestinal irritation beyond bioavailability enhancement [73]. Villanueva-Martínez et al. [74] developed two nano-systems (microemulsions and nanosuspensions) for transdermal delivery of sodium alendronate. In vivo, tests showed that neither microemulsions nor nanosuspensions exhibited skin irritation. They evaluated the permeability of two formulations through the pig skin and found that the permeability of microemulsions was preferable to that of the nanosuspension. Therefore, alendronate-loaded microemulsions were finalized as a viable transdermal nanocarrier for OP therapy.

### 3.4. Scaffolds Implantation and Local Drug Delivery

Physically or chemically conjugating drugs to scaffolds can achieve local delivery of anti-OP drugs and systematic treatment while repairing the bone defects [75]. Topical application can reduce the dose administered and reduce the side effects due to high systemic exposure, thus improving drug safety [76]. Considering these merits, Sun et al. [77] loaded strontium ranelate onto the surface of a porous sulfonated polyetheretherketone scaffold. In vitro cell assay results showed that the sulfonated polyetheretherketone scaffold containing strontium ranelate enhanced the adhesion ability of MC3T3-E1 cells and improved the mineralization and deposition of extracellular matrix mineralization. Chiang et al. [78] encapsulated strontium ranelate into nanoparticles using glycol chitosan with hyaluronic acid and then loaded these nanoparticles into polyethylene glycol diacrylate hydrogels. Studies have shown that the medicament had no cytotoxic effect on osteoblasts in vitro while presenting a certain ability to induce bone regeneration in vivo. The above findings indicate that scaffold implantation is also an effective strategy for delivering anti-OP drugs, which is expected to improve efficiency and depress toxicity. Bone tissue engineering as a novel delivery strategy has the advantages of improving drug targeting and achieving sustained drug delivery. However, bone tissue engineering suffers from high manufacturing costs, abrupt release, risk of infection at the site of action, difficulty in obtaining regulatory approval, and clinical trials that have not yet been conducted on a large scale. Bone tissue engineering requires more in-depth research to select appropriate preparation methods to minimize toxic residues, improve cost-effectiveness, and improve drug-carrying technologies to optimize release kinetics. In addition, improved synchronization of regulatory and scientific research and attention to regulatory harmonization of biomedical devices on a global scale will accelerate the pace of commercialization of bone tissue engineering [79,80].

## 4. Delivery Vehicles and Strategies for Anti-OP Drugs

Based on the above drug delivery ideas, the current anti-OP drug delivery strategies mainly include micro/nano drug delivery systems, bone tissue engineering, and microneedles. For poorly water-soluble drugs, emulsions, micelles, and other supersaturated drug delivery systems can be used to solubilize them, and the surface modification of nanocarriers can realize the targeted delivery of drugs and achieve the effect of increasing efficiency and reducing toxicity [81]. The following section summarizes the common strategies for OP drug delivery and compares the advantages and limitations of several delivery strategies (see Table 2).

### 4.1. Hydroxyapatite Nanoparticles

Hydroxyapatite nanoparticles (HA-NPs) exhibit stable mechanical performance in the physiological environment and can support the attachment and growth of bone cells. HA-NPs also provide crystal nuclei during osteocyte calcification, exerting bone conduction [90]. Fouad-Elhady et al. [91] explored the anti-osteoporotic effects of HA-NPs, chitosan/HA-NPs (nCh/HA-NPs), and silver/HA-NPs (nAg/HA-NPs). nCh/HA-NPs had good biocompatibility and bone conductivity, while nAg/HA-NPs presented good antimicrobial properties and in vitro bone induction potential [92]. All three nanoparticles can reduce serum bone alkaline phosphatase and salivary protein levels, which explicitly affect bone resorption resistance. Therefore, HA-NPs have the potential to deliver anti-OP drugs to the bone. Marycz et al. [93] prepared composite nanoparticles (nHAp/IO@APTES) composed of HA-NPs, iron oxide, and (3-aminopropyl) triethoxysilane (APTES). It was shown that nHAp/IO@APTES increased the viability of osteoblasts, promoted the expression of Runx2, reduced the activity of key pro-inflammatory cytokines, and inhibited the activity of osteoclasts. The modification of APTES increased the biocompatibility of nanoparticles, and the presence of iron oxide allowed the nanoparticles to release drugs in response to external stimuli. This study demonstrated that nHAp/IO@APTES as an anti-OP drug delivery system works well.

Wu et al. [94] loaded simvastatin into mesoporous HA-NPs, on which the surface was pre-modified with poly (*N*-isopropyl acrylamide). Poly(*N*-isopropyl acrylamide) is a heat-responsive polymer whose chains exhibit an elongated conformation in water that can encapsulate the drug into mesopores under temperatures below 32 °C, while above 32 °C, poly(*N*-isopropyl acrylamide) collapses in a hydrophobic state on the surface of mesoporous particles to achieve slow release of simvastatin [95]. Mesoporous HA-NPs significantly promoted osteogenic differentiation of BMSCs, improved bone formation in OVX rats, and inhibited osteoclast activity. The responsive release system can realize controlled anti-OP drug release and change payloads’ release profile. Kotak et al. [96] developed a bone-targeted delivery system for calcitonin using HA-NPs as carrier materials. Calcitonin was loaded on HA-NPs by ion complexation. The resulting calcitonin-laden HA-NPs had a high drug-loading rate (about 85%). The relative bioavailability of calcitonin HA-NPs administered via the sublingual mucosa was approximately 15% compared with subcutaneous injection. The results suggest that HA-NPs offer the opportunity to deliver calcitonin sublingually, dramatically improving the patient’s medication compliance.

### 4.2. Liposomes

Liposomes have a biofilm-like lipid bilayer vesicular structure composed of phospholipids and cholesterol with an average particle size of tens to hundreds of nanometers [83]. Phospholipids consist of hydrophilic and hydrophobic tails that form self-assembled nanostructures through a hydrophobic effect [97]. Hydrophilic molecules can be encapsulated in the inner aqueous phase, and lipophilic molecules can be entrapped into the lipid bilayers [98]. Liposomes possess the advantages of superior biocompatibility, low immunogenicity, increased drug stability, prolonged drug half-life in vivo, targeted drug delivery, etc., and are frequently employed as excellent carriers for delivery of various drugs [99].

Cai et al. [100] prepared HA-modified liposomes loading zoledronic acid. Zoledronic acid was encapsulated in the hydrophilic core of the liposomes. Since HA is negatively charged, it can electrostatically interact with the surface of positively charged lipid molecules, thus forming a coating on the surface of liposomes [101,102]. The lipid carrier allowed the slow release of zoledronic acid at the target site, acting as a drug reservoir. The modification of HA enabled liposomes to be deposited into the bone tissue. At the same time, zoledronic acid had a strong affinity for HA in the bone, further enhancing its efficacy in treating OP. It was concluded that a ligand-binding liposomal system could be suitable for systemic delivery of anti-OP drugs. Alendronate-tris (ethylene glycol)-cholesterol conjugate (ALN-TEG-Chol) has been reported to have a strong binding capacity for HA [103]. Based on this point, Sun et al. [104] designed and synthesized a novel pyrophosphate-tris (ethylene glycol)-cholesterol conjugate (PPi-TEG-Chol) and modified it onto the surface of icariin-loaded liposomes as a targeting moiety. The experimental results showed that targeted liposomes increased bone strength in OVX rats and inhibited bone resorption to a certain extent. Icariin-loaded HA-targeted liposomes may be a promising candidate for OP therapy.

Depending on high convenience and compliance, pharmacists have also developed liposomes for oral drug delivery [105]. Gradauer et al. [106] conjugated thioglycolic acid grafted chitosan to 6,6′-dithionicotinamide and decorated calcitonin liposomes with modified chitosan. Chitosan loosened the tight junctions between intestinal epithelial cells, thus increasing the oral bioavailability of calcitonin by 8.2 times through liposomes compared to that of free calcitonin solution. This study shows that nanoencapsulation of peptides can protect them from degradation during transportation across harsh gastrointestinal environments.

Positively charged cationic liposomes are more likely to bind to negatively charged nucleic acids to form complexes with high affinity for cell membranes [107]. Hu et al. [108] encapsulated anti-mir-132 in (AspSerSer)_6_-modified cationic liposomes. The modification of (AspSerSer)_6_ endowed liposomes with a bone-targeting ability. The results showed that liposomes could successfully target the bone and silence the expression of miRNA-132-3p in BMSCs to reverse OP. Therefore, it was concluded that liposomes could be invoked as a vehicle for gene delivery to treat OP.

### 4.3. Emulsions

Tocotrienols exert their osteoprotective effects as free radical scavengers on cell membranes against oxidative stress due to estrogen deficiency [109]. However, the oral bioavailability of tocotrienols is low. To ameliorate the oral bioavailability of tocotrienols, Mohamad et al. [110] delivered tocotrienols using a self-emulsifying drug delivery system (SEDDS). SEDDS was composed of Cremophor^®^ EL (International Lab, San Francisco, CA, USA), Labrasol^®^ (International Lab, San Francisco, CA, USA), Captex^®^ 355 (International Lab, San Francisco, CA, USA), and corn oil. Emulsifiers could increase the solubility of tocotrienols in the formulation during emulsification, thereby facilitating oral absorption. The plasma levels of δ-tocopherols and antioxidant enzymes in the SEDDS group were significantly higher than those in the free drug group. SEDDS improved the cortical bone thickness and bone strength of OVX rats. Thus, emulsions can potentiate tocotrienols to treat OP.

The solubility of raloxifene in water is poor, compromising its therapeutic efficacy. O/W nanoemulsions can effectively increase the solubility of raloxifene in aqueous medium [111]. Zakir et al. [112] developed raloxifene nanoemulsions. They loaded them into a hydrogel composed of poloxamer 407 and carbomer 934 to prolong the adhesion time of milk droplets on the nasal mucosa. The emulsions consisted of Labrafac™ WL 1349, Labrasol^®^, Cremophor^®^ EL, Labrafac™ PG, Capryol™ 90, Lauroglycol™ 90, Transcutol^®^ P, Transcutol^®^ HP, and Peceol™ (Gattefosse India Pvt Ltd., Mumbai, India). The bioavailability of raloxifene in nanoemulgels was 7.4 times higher than that of commercially available tablets. Compared to oral tablets, the in vivo pharmacodynamic studies showed a 162% increase in bone density in rabbits in the nanoemulgel group. These findings reveal that nanoemulgels may be an effective delivery system for raloxifene to treat OP.

Guo et al. [113] prepared teriparatide microemulsions for oral administration. The microemulsions consisted of Crodamol^®^ GTCC (Gattefosse China Trading Co., Ltd., Beijing, China), nonionic surfactants Labrasol^®^ (Beijing Fengli Jingqiu Commerce and Trade Co., Ltd., Beijing, China), permeability enhancer Solutol^®^ HS 15 (Beijing Fengli Jingqiu Commerce and Trade Co., Ltd., Beijing, China), D-α-tocopheryl acetate, and normal saline as aqueous phase. The loading rate of microemulsions was 85%, the bioavailability via oral administration was 5.4%, and it was 12.0% via ileal injection. After treatment for 8 weeks by gavage at a dose of 0.05 mg/kg, the proximal tibial bone mineral content and bone density of OVX rats increased by 11.0% and 6.4%, respectively. The above findings indicate that the inclusion of teriparatide in microemulsions can inhibit its degradation by proteases and increase the gastrointestinal absorption of teriparatide. Thus, oral microemulsions may be a potentially effective approach to delivering teriparatide for OP treatment.

### 4.4. Dendrimers

Dendrimers are spherical and highly branched polymers with a broad cavity structure and a dense surface of active functional groups, empowering dendrimers to carry interior and outer drugs. The particle size, surface charge, and functional groups of dendrimers can be finely tuned during synthesis. In addition to encapsulating the drug in the internal cavity, functional groups outside the dendrimers can covalently bind to suitable drug molecules to enable drug delivery [114].

Yamashita et al. [47] developed a third-generation polyamide amine (PAMAM) dendritic polymer conjugated with polyethylene glycol (PEG). The PAMAM backbone was further conjugated with four kinds of amino acids: aspartic acid (Asp), glutamic acid (Glu), succinic acid (Suc), or aconitum acid (Aco), respectively, to obtain four different types of modified PAMAMs. The four amino acids used above are rich in carboxylic acids, which can increase the affinity of PAMAM to HA. Tissue distribution studies were conducted after intravenous injection to mice, showing a bone deposition of 11.3% for PAMAMs. In contrast, the bone deposition of PEG-Asp-PAMAM, PEG-Glu-PAMAM, PEG-Suc-PAMAM, and PEG-Aco-PAMAM was approximately 46.0%, 15.6%, 22.6%, and 24.5%, respectively. The above facts suggest that PEG-ASP-PAMAM has the potential to serve as a drug carrier for targeted delivery of anti-OP drugs.

Bone-targeted delivery is typically achieved using molecules that have a high affinity to the inorganic matrix of the bone or that are highly selective for bone-associated cells. Ren et al. [115] developed a CH6 aptamer and C11 peptide dual-targeted delivery system (CH6-PAMAM-C11) based on PAMAM dendrimers for vitamin D delivery to osteoblasts. C11 peptides derived from 11 amino acids at the enamelin’s C-terminus can regulate HA formation and have strong interactions with HA [116]. Since PAMAM dendrimers have many positively charged amino groups with certain cytotoxicity [117], Ren et al. blocked the remaining amino groups with shorter methoxylated PEG after C11 peptides. Meanwhile, CH6 aptamers were bound to PAMAM by heterofunctional double-substituted PEG, which reduced the toxicity of the nanocarrier and improved its biocompatibility. CH6-PAMAM-C11 had a diameter of 40~50 nm, showing a remarkable ability to carry small molecule drugs. After administration, CH6-PAMAM-C11 was successfully targeted and accumulated in the bone within 24 h and then delivered the drug to the target site of osteoblasts. More accumulation of dual-targeted nanocarriers (CH6-PAMAM-C11) than single-targeted nanocarriers (PAMAM-C11 and PAMAM-CH6) was investigated in the periosteal layer of rat skulls. The targeting efficiency of dual-targeted nanocarrier was significantly better than single-targeted carriers, and no off-target effects were observed in vital organs other than kidneys. The above results indicate that CH6-PAMAM-C11 may be a promising nanocarrier for bone-specific delivery of anti-OP drugs.

### 4.5. Micelles

Micelles consist of a hydrophobic core and a hydrophilic shell. When amphiphilic molecules in an aqueous solution reach a critical concentration, the molecules can self-assemble into ordered thermodynamically stable nanoparticles, known as micelles [118]. Micelles hold great promise in the field of drug delivery. Micelles have the advantages of small particle size, low toxicity, simple preparation, and high feasibility of scale-up production. Furthermore, micelles can improve the solubility of poorly soluble drugs and be used as carriers of both small-molecule and biotech drugs. In addition, surface-modified micelles can also be endowed with good targeting ability [119].

Based on the merits above, some scholars have tried to deliver miR214 antagonists to bone to treat OP. However, miRNA delivery often lacks tissue/cell specificity and is easily degraded by nucleases [120]. Cai et al. [121] developed polyurethane micelles modified by Asp_8_ to address the challenges in miRNA delivery. Polyurethane consists of multiple urethane units and has good biocompatibility, low cytotoxicity, and good flexibility, which can improve the stability of encapsulated drugs [49]. Polyurethane micelles encapsulate miRNAs by electrostatic interaction. Serum stability examination and cytotoxicity assay showed that polyurethane could protect miRNAs from endogenous nuclease degradation and good biocompatibility was provided with Asp_8_ and polyurethane carrier. Inflammatory cytokines such as IL2, IL6, TNFα, and IFNγ are involved in osteoclast production through transcription factors that regulate osteoclasts positively or negatively [122,123]. Quantitative analysis by enzyme-linked immunosorbent technique showed that serum TNFα, IFNγ, IL2, and IL6 levels were unaffected, suggesting that the downregulation of osteoclast activity was ascribed to the specific delivery of miR214 antagonists by Asp_8_-polyurethane micelles, rather than from an inflammatory response. After injection of micelles containing miR214 antagonists, osteoclast-associated genes (TRAP and CTSK) were downregulated and bone mass in OVX mice was significantly increased. In another study, Sun et al. [48] developed SDSSD-modified polyurethane micelles for osseous delivery of miR214 antagonists. Ameliorative therapeutic outcomes were also achieved through such nanocarriers. Therefore, polyurethane micelles represent an excellent vehicle for specifically delivering gene drugs to treat OP.

Xie et al. [124] delivered atorvastatin to the bone tissue using PEG-PLGA micelles. They attached tetracycline molecules to micelles to obtain TC-PEG-PLGA micelles. TC-PEG-PLGA micelles exhibited low cytotoxicity, with more than 80% of MC3T3-E1 cells surviving at a concentration of up to 600 μg/mL. The in vitro binding rate of TC-PEG-PLGA micelles to HA was 77.3%. Regarding the pharmacodynamic study, bone strength in the TC-PEG-PLGA micelles group significantly increased compared with the free atorvastatin and PEG-PLGA micelles groups. The results showed that TC-PEG-PLGA micelles demonstrate significant prospects in targeted therapy of OP. Inspired by these encouraging outcomes, Xie et al. [125] further examined the bone-targeted delivery performance of this vehicle using simvastatin as a therapy drug. Micelles prolonged the systemic circulation time of simvastatin and preferentially accumulated in the bone tissue. TC-PEG-PLGA micelles containing simvastatin increased bone density and improved bone strength in OVX rats. It can be inferred that micelles are also an excellent vehicle for bone-targeted drug delivery.

### 4.6. Other Polymeric Nanoparticles

Fazil et al. [126] optimized PLGA nanoparticles loading risedronate based on the Box-Behnken design of the experiment. They evaluated the ex vivo permeability of formulation using the porcine nasal mucosa, and the accumulative permeability of the encapsulated drug was measured to be 34.32 ± 2.64%, significantly higher than that of the free counterpart. Jing et al. [127] developed PLGA nanoparticles with chitosan or alendronate modification as vehicles for bone-targeted drug delivery. The nanoparticles displayed sustained release without apparent burst release and good cytocompatibility against MC3T3 cells, an osteoblastic cell line. Alendronate-modified nanoparticles showed a high affinity to HA, illustrating their possibility for bone-targeted drug delivery. Unlike nanoparticles without alendronate modification, alendronate-modified nanoparticles were preferentially taken up by MC3T3 cells. The alendronate-modified nanoparticles may be developed as a bone-targeted vehicle for OP treatment.

Santhosh et al. [128] developed deacetylated chitosan nanoparticles containing risedronate. Due to the negative charge of risedronate and the positive charge of chitosan, nanoparticles’ encapsulation rate and drug loading were fairly high. In the pharmacodynamic studies, the bone density of the rats in the treatment group was significantly improved, the microstructure of the trabecular bone was significantly improved, and the porosity of cortical bone was narrowed. Oral administration of risedronate is of gastrointestinal irritation. Using chitosan nanoparticles increased the bioavailability of risedronate and reduced the side effects associated with oral administration.

Although chitosan can bind to DNA and protect it from nuclease degradation, it is less efficient in transfection [129]. Known for its “proton sponge effect”, polyethyleneimine (PEI) is one of the most common nonviral gene vectors, but it is highly cytotoxic and biodegradable [130]. To this end, Zhao et al. [131] utilized chitosan and 1.8 kDa PEI to deliver BMP2. When the mass ratio of chitosan to PEI was 20:1, the prepared chitosan-PEI/BMP2 nanoparticles retained the advantages of low chitosan cytotoxicity and had higher transfection efficiency. The mineralization of MC3T3-E1 cells treated with chitosan-PEI/BMP2 nanoparticles was remarkable. After 12 weeks of administration, chitosan-PEI/BMP2 nanoparticles significantly increased new bone formation in rats with bone defects, suggesting that chitosan-PEI/BMP2 nanoparticles have potential applications in the treatment of bone defects in the future, which may be helpful in the delivery of anti-OP drugs for OP therapy.

Nano-drug delivery systems have a proven track record in drug delivery and therapy. Likewise, they play a prominent role in the intervention against osteoporosis. Table 3 lists the reported nanocarrier systems for osteoporotic drug delivery and treatment.

### 4.7. Bone Tissue Engineering Scaffolds

Bone tissue engineering scaffolds have a biomimetic structure like natural bone, which can provide cells with the three-dimensional space needed for survival so that cells can obtain sufficient nutrients. The scaffold material is gradually degraded while the planted bone cells continue to proliferate and differentiate, inducing new functional bone regeneration to facilitate the repair of bone tissue defects [132]. Ideal bone tissue engineering scaffolds should be qualified with the following attributes [133,134,135]: (1) having excellent biocompatibility and not causing immune rejection or other toxic reactions in the body; (2) possessing a certain three-dimensional shape and good mechanical strength able to maintain the original shape in the body; (3) having large porosity and surface area conducive to cell attachment and providing a suitable microenvironment for cell proliferation and differentiation; (4) presenting an acceptable biodegradation rate consistent with the growth rate of bone tissue. Figure 4 illustrates the bioprocess of bone tissue engineering scaffolds in treating osteoporotic bone defects.

Currently, the commonly used materials in the preparation of bone tissue engineering scaffolds are mainly natural polymer materials (e.g., collagen, gelatin, chitosan, fibrin, and hyaluronic acid), synthetic biological materials (e.g., polylactic acid (PLA), polyglycolic acid (PGA), (PLGA), tricalcium phosphate, and bioactive ceramics), and metal materials (tantalum, titanium, magnesium, etc.) [136,137]. Composite materials stand as one of the research hotspots. The scaffolds prepared from single-component materials have apparent limitations and often cannot meet the requirements for excellent bone tissue engineering scaffolds. Composite materials integrate the advantages of the two original materials, complementing their strengths and weaknesses [138]. By adjusting the proportion of each material, the treatment requirements for different diseases can be satisfied [78]. For example, Rasoul et al. [139] incorporated magnesium particles into PLA matrix to prepare PLA-Mg composite scaffolds, which improved the shortcomings of poor mechanical properties and slow degradation rate of PLA. Compared with pure PLA, PLA-Mg composite scaffolds have better mechanical properties, higher cell compatibility, and more suitable degradation behavior.

Physical embedding of microspheres or nanoparticles into scaffolds can systematically treat OP while repairing bone defects [140]. Che et al. [141] loaded heat-sensitive teriparatide liposomes into polydopamine-coated mesoporous bioactive glasses. Polydopamine converted light into heat under near-infrared irradiation. When the temperature was raised above the phase transition temperature, the extravasation of heat-sensitive liposomes would be significantly enhanced, resulting in rapid drug release. Once the temperature was below the phase transition temperature, liposomes quickly returned to a dense structure with little or no drug release. This heat-responsive drug delivery system enabled precise pulse release of teriparatide. After administration, the composite scaffolds resulted in strong bone tissue regeneration in the defect center of rats with skull defects. In another study, Kuang et al. [142] loaded heat-sensitive teriparatide microspheres into a hydrogel scaffold with indocyanine green as a photothermal response material. Under near-infrared irradiation, the microspheres decomposed, followed by drug release. The findings above suggest that the physical combination of micro/nanoparticles or active drugs with scaffolds can treat bone defects and enable local delivery and controlled release of active drugs.

In addition to physical combination, active drugs, microspheres or nanoparticles can be chemically conjugated with scaffold materials, but this approach is only suitable for some specific drugs and scaffold materials. Cholesterol-modified non-coding miRNAs (Chol-miR-26a) can promote osteogenic differentiation of human bone marrow mesenchymal stem cells. Gan et al. [143] bound Chol-miR-26a to injectable PEG hydrogel scaffolds via UV-cleavage ester bonds. Under UV irradiation, Chol-miR-26a could be released from the gel networks, enabling precise and controlled release of miR-26a. Zhang et al. [144] prepared another hydrogel scaffold. They first grafted dopamine (DA) to alginate (Alg) by an amide reaction to obtain Alg-DA, and then mixed strontium ions with Alg-DA aqueous solution where the carboxyl group of Alg-DA can form dynamic ionic bonds with strontium ions. The catechol groups on Alg-DA allowed the hydrogels to form stable chemical crosslinks while giving the hydrogel good tissue adhesion. Therefore, the specific binding between the drug and the scaffolds provides an innovative way to deliver anti-OP drugs.

In recent years, computer-aided 3D-printing technology has become a popular method for scaffold production. This technology can process and modify the scaffold’s surface structure and overall shape more finely, making it easier to translate a scaffold into a drug delivery system [145]. Zou et al. [146] embedded PLGA microspheres containing icariin into 3D-printed polycaprolactone/nano-HA composite scaffolds. Four weeks after the implantation of scaffolds, many collagen fibers and connective tissue accumulation were observed at the skull defect site. Furthermore, the skull defect area of the scaffold group was smaller than that of the blank group, indicating that the composite scaffolds had better bone regeneration ability. Chen et al. [147] utilized 3D-printing technology to make a loofah-like HA scaffold. The biomimetic structure of the natural loofah allowed the scaffold to have high porosity and connectivity. The loofah-like scaffold showed good osteogenic performance in the rabbit skull defect model. With the increasing maturity of 3D-printing technology, the scaffolds can ideally modulate the release curve of drugs, suggesting 3D-printing technology provides a new method for developing precision micromachined bone tissue engineering scaffolds.

### 4.8. Microneedles

Since the first microneedle patent application was passed in the 1970s, studies on microneedles for drug delivery continue to grow in popularity. Microneedles, consisting of multiple micron-sized tiny tips attached to the base in an array, are an emerging physical osmotic technology [148]. Microneedles can penetrate the stratum corneum and form micron-sized channels, importing active drugs directly into the epidermis or upper dermis [89]. Controlling the length of the microneedle avoids contact with capillaries and nerve endings, thus reducing the patient’s pain during administration [149]. For types of microneedles and preparation techniques see [71,150]. As shown in Figure 5, solid microneedles, coated microneedles, hollow microneedles, and soluble microneedles have been preferentially developed by practitioners to deliver anti-OP drugs [151]. Table 4 summarizes the microneedles used to deliver anti-OP drugs.

#### 4.8.1. Solid Microneedles

Solid microneedles are typically made of silicon, non-degradable polymers, or metals. This type of microneedle itself does not carry drugs, and its role is to puncture the epidermis to form micropores required for drug penetration. The pre-implanted microneedles are removed upon medication, followed by drug attachment to the microneedle puncture site. Along the pre-formed micropores, the drug passively penetrates the skin [163]. Jyoung et al. [152] delivered alendronate sodium using solid microneedles and iontophoresis. Compared with administration by iontophoresis, microneedle pretreatment combined with iontophoresis increased the amount of drug penetrating the skin of hairless mice by a factor of 70. Solid microneedles are the earliest developed modality, and the shortcomings are also obvious, e.g., the possibility of needles breaking. Broken needles trapped in the body will bring safety concerns to patients. In addition, microwells produce different dynamic recovery processes due to individual differences, making it hard to precisely control the drug release rate and final dose [164].

#### 4.8.2. Coated Microneedles

Coated microneedles refer to a drug delivery system that attaches drugs to the surface of microneedles by infiltration, coating, etc. [165]. The dose administered can be controlled by drug concentration, coating thickness, density and number of microneedles, and insertion time [166]. Currently marketed calcitonin is administered by intramuscular injection, subcutaneous injection, and intranasal administration. However, injection has the problems of needle infection and side effects due to high blood concentration, and intranasal administration also faces the side effects of irritation of the nasal mucosa leading to rhinitis, and the bioavailability of the drug is only 3%. Tas et al. [153] developed a microneedle patch of calcitonin using coated microneedles. In vivo studies conducted on hairless rats showed no significant difference in the bioavailability of coated microneedles compared to subcutaneous injections. However, the microneedle system resulted in 13 times more bioavailability than nasal sprays. This study suggests that microneedle delivery may be a viable alternative strategy to calcitonin delivery. Oh et al. [154] prepared two teriparatide microneedles with different dissolution rates using sucrose and sodium carboxymethyl cellulose (CMC-Na), respectively. In the in vitro dissolution test, it took 30 min for CMC microneedles to release 80% of teriparatide, while it was 10 min for sucrose microneedles. This suggested that sucrose-coated microneedles dissolve quickly, while CMC-Na-coated ones could sustain the drug release. After medication, the maximum plasma drug concentrations of CMC-Na-based and sucrose-based microneedles were 868 pg/mL and 6809 pg/mL, and those of the AUC values were up to 6771 pg·hr/mL and 17,171 pg·hr/mL, respectively. This study provides a methodology for optimizing the pharmacokinetic parameters of anti-OP drugs using coated microneedles through a transdermal route. In the future, microneedles may be a viable option for peptide drugs delivery to treat OP.

#### 4.8.3. Dissolving Microneedles

Biodegradable polymeric materials commonly used in the preparation of dissolving microneedles include carboxymethylcellulose, starch, polyvinylpyrrolidone, PLGA, chitosan, cyclodextrin, and sodium alginate [167]. After the dissolving microneedles are pierced into the skin, the needle body will gradually degrade in the microenvironment, and the drug will be released synchronously, followed by absorption into the human body through the subcutaneous tissue [168]. Dissolving microneedles can completely dissolve in the skin after application, thus overcoming the defect of solid microneedle breakage. NAITO et al. developed dissolving microneedles composed of hyaluronic acid loaded with human parathyroid hormone (1–34) (PTH). Plasma concentrations of PTH increased rapidly after microneedle administration in rats, with a relative bioavailability of 100 ± 4% compared to subcutaneous injection. The results indicated that the rat OP model of PTH microneedles could effectively inhibit the reduction of bone density and had little skin irritation [159]. Katsumi et al. [169] loaded alendronate sodium into hyaluronic acid microneedles and 75% of needles dissolved within 30 min. The plasma concentration of alendronate sodium increased rapidly after application to the skin, resulting in a bioavailability of approximately 90%. However, mild erythema appeared on the skin of the rats after 4 days of administration. Katsumi et al. [157] speculated that this may be caused by sodium alendronate remaining in the skin at the bottom of the needles. Therefore, they improved the microneedles by dipping only alendronate sodium on the tips of the micron-sized needles. After 5 min of application on the rat skin, the microneedles’ tips dissolved completely, resulting in approximately 96% bioavailability of alendronate sodium. It was assumed that impregnating alendronate only in the tips of the needles might allow the drug to be released only in the dermis layer and be rapidly absorbed, thereby reducing skin damage caused by alendronate microneedles. In addition, loading alendronate sodium on the tips of the needles increased the bioavailability of the drug. Thus, delivery of bisphosphonates by microneedles can reduce the irritating side effects of oral bisphosphonates and improve their bioavailability and treatment efficacy.

#### 4.8.4. Hollow Microneedles

Hollow microneedles are “small syringes” arranged in a reduced form and are driven by controlled pressure to deliver drugs into the skin [170,171]. Hollow microneedles are mainly used for loading proteins, oligonucleotides, and other macromolecular drugs [89]. Compared with other methods, hollow microneedles have the advantages of fast drug delivery, are painless, and allowing the use of larger drug doses [172,173]. However, hollow microneedles risk needle blockage by skin tissue, which can be avoided by placing the pinhole sideways [174]. Bushra et al. [162] used 3D-printed hollow microneedles to deliver denosumab. In vivo studies showed that hollow microneedles released the drug at a rate similar to that in the subcutaneous group and did not cause painful stimuli. This study has significant implications for the treatment of osteoporosis, offering patients a new treatment option.

## 5. Anti-OP Preparations in Clinical Trials

Thanks to the continuous efforts of pharmacists, many anti-OP formulations that take full advantage of newly developed delivery technologies or formulation strategies have entered the clinical stage. Table 5 lists some anti-OP preparations based on novel preparative techniques subjected to clinical investigation or abortion.

Tarsa Therapeutics holds a patent for a co-formulation of citric acid and calcitonin. The citric acid in this enteric-coated formulation is used to lower the pH of the intestine to reduce protease activity. Still, citric acid does not significantly increase the permeability of the drug in the intestine. In the results of the phase III clinical trial (NCT00959764), there was no difference in bone mineral density between the formulation and placebo after administration, so FDA did not approve the new drug application [175]. GlaxoSmithKline conducted a phase II clinical trial of oral rhPTH(1–31)NH_2_ (NCT01321723) [176]. This oral tablet contains 5 mg of active drug, coated citric acid granules (as pH adjuster), and lauroyl or palmitoyl carnitine. After 6 months of medication, there was a significant increase in the lumbar spine bone density, but no significant correlation existed between the investigated dose and adverse effects. Novartis conducted a phase III clinical trial (NCT00525798) for the drug SMC021, in which 8-[(5-chloro-2-hydroxybenzoyl) amino] octanoic acid (5-CNAC) was expected to improve the oral bioavailability of the active drug calcitonin [177]. However, the absorption of calcitonin in the gastrointestinal tract was insufficient and could not achieve the effect of preventing fractures; also, both the safety and efficacy of SMC021 were disappointing after a full assessment. Entera Bio’s oral teriparatide (EBP05) completed a phase II clinical trial (NCT04003467) in 2021. After six months, participants receiving a 2.5 mg dose of EBP05 showed significant dose-related increases in the bone mineral density in the lumbar spine, femoral neck, and bone and hip joints [178]. P1NP was significantly increased in the 2.5 mg dose group compared to the placebo group (*p* < 0.05). P1NP is a biomarker that indicates the rate of new bone formation. The safety profile of EBP05 was consistent with that of the injectable agent (Forteo^®^) on the market, and there were no serious adverse effects related to the drug. Currently, Entera Bio is planning an EBP05 Phase III registration study.

Zosano Pharma, in collaboration with Eli Lilly, developed a novel coated microneedle delivery system (ZP-PTH) for transdermal delivery of teriparatide. In a phase II clinical trial (NCT01011556), after 24 weeks of treatment, participants’ lumbar spine bone density was significantly improved, and the increase in the lumbar spine bone density was proportional to the dose [179]. Three doses of ZP-PTH (20 μg, 30 μg, and 40 μg) increased the lumbar spine bone density by 3.0%, 3.5%, and 5.0%, respectively. The curative effect was better than the placebo control’s (*p* < 0.001). Compared with marketed injections, *T*_max_ was shorter (0.12 h ± 0.05) and *C*_max_ was higher (1.7 times higher) in the 40 μg ZP-PTH group. All trial ZP-PTH doses were well tolerated, and participants experienced no systemic adverse effects during administration. Based on the phase II clinical trial results, Eli Lilly is ready to advance to a phase III clinical trial. In addition, abaloparatide-coated microneedles were jointly developed by Radius and Kindeva. The transdermal drug delivery system loading abaloparatide (abalo-sMTS) has also entered the phase III clinical trial (NCT04064411).

## 6. Perspectives

Lower levels of vascularization in the bone, low blood flow, and the blood–bone marrow barrier formed by lining cells presents significant challenges to anti-OP drug delivery compared to that for other tissues [180]. The physiological barriers require a high administration frequency and dose of anti-OP drugs, which tend to bring about potential side effects to patients. In addition, anti-OP drugs generally require long-term medication. Although many preclinical studies have made considerable progress, overall evaluation of the long-term safety of nano-formulation is needed. High bioavailability and marginal adverse effects should be important criteria for determining an anti-OP drug and its delivery systems. Various nanocarriers show a magnificent potential in drug delivery application. Still, the nanoparticles ingeniously designed in preclinical studies may be difficult to scale up for mass production, and their unique properties, such as particle size, surface potential, and cell/tissue affinity, may vary largely. With the advancement in nanotechnology, more and more new materials and technologies have been applied to the engineering of nanoparticles. It is foreseeable that in the upcoming years, smarter drug delivery systems will break through these limitations and step into clinical investigation to overcome the dilemmas of osteoporosis treatment. Expressly, transdermal drug delivery systems such as microneedles have unparalleled advantages compared with other systems in safety and compliance, so they may win a place in anti-OP therapy.

## 7. Conclusions

This work systemically reviews the pathogenesis of OP, the engineering of anti-OP drug delivery, novel anti-OP treatment strategies, and clinical anti-OP drug preparations, and especially elaborates several promising drug delivery and therapy systems, including HA-NPs, liposomes, emulsions, dendrimers, micelles, bone tissue engineering scaffolds, and microneedles. These delivery systems and therapy strategies intended for bone targeting effectively optimize drug delivery efficiency and therapeutic outcomes and alleviate side effects. They are expected to replace the traditional invasive drug delivery mode to cure osteoporosis. Existing clinical OP treatment protocols are inevitably flawed in one way or another. The vigorous development of these emerging delivery systems shows promise to solve these vexing problems, such as low bioavailability, insufficient osseous drug exposure, and frequent dosing. Nevertheless, ongoing efforts are highly inspired to translate micro/nanomedicines from the bench to the clinic.

## Figures and Tables

**Figure 1 molecules-28-06652-f001:**
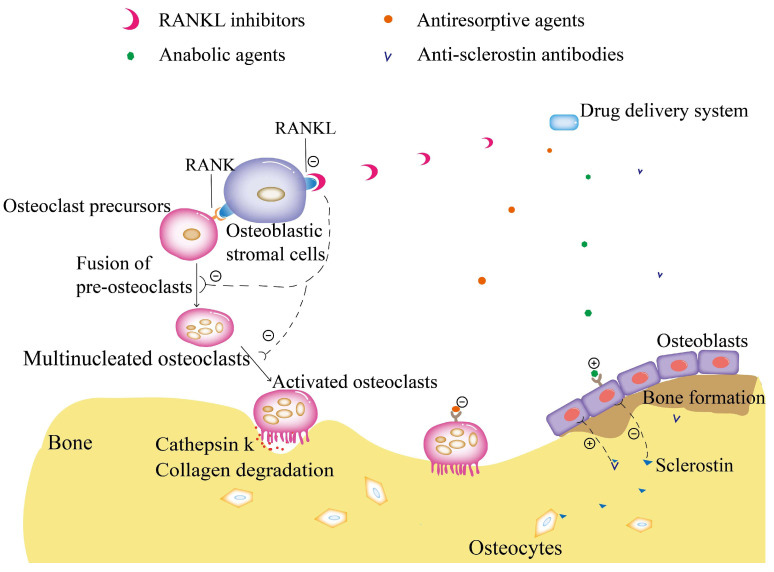
Different anti-OP drugs and their mechanisms of action on specific cell types in the bone. (RANK: receptor activators for nuclear factor κ-B; RANKL: receptor activator of nuclear factor κ-B ligand).

**Figure 2 molecules-28-06652-f002:**
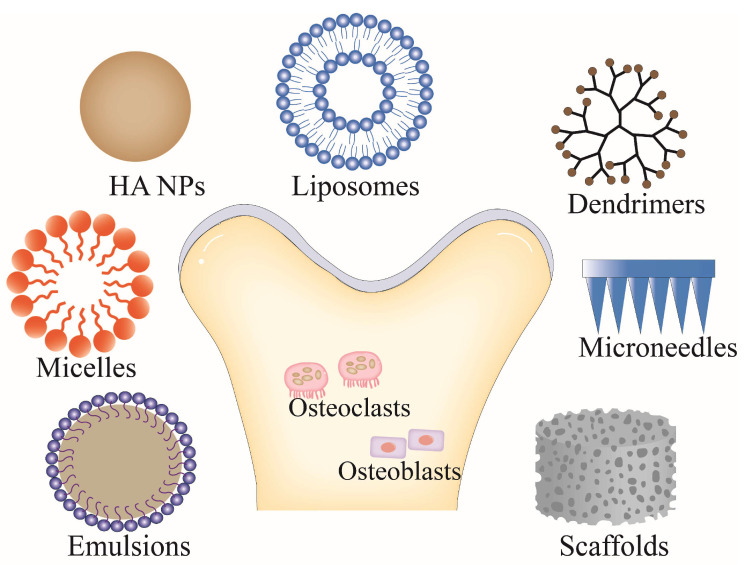
Novel drug delivery systems for OP treatment. (HA NPs: hydroxyapatite nanoparticles).

**Figure 3 molecules-28-06652-f003:**
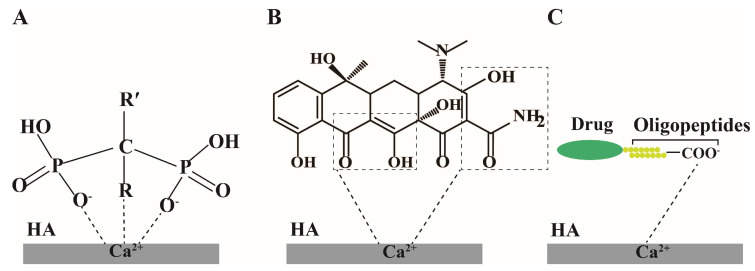
Bisphosphonates, tetracycline, and oligopeptides as bone-targeting ligands for HA modification. ((**A**): bisphosphonates bind to Ca^2+^ in HA; (**B**): tetracycline binds to Ca^2+^ in HA; (**C**): oligopeptides bind to Ca^2+^ in HA).

**Figure 4 molecules-28-06652-f004:**
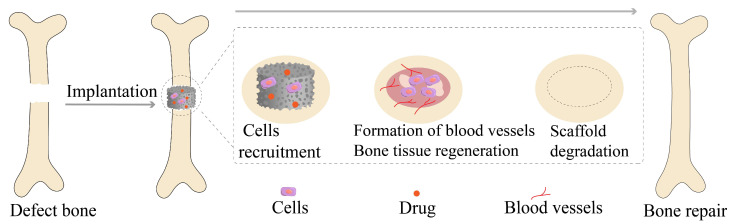
The bioprocess of bone tissue engineering scaffolds treating osteoporotic bone defects.

**Figure 5 molecules-28-06652-f005:**
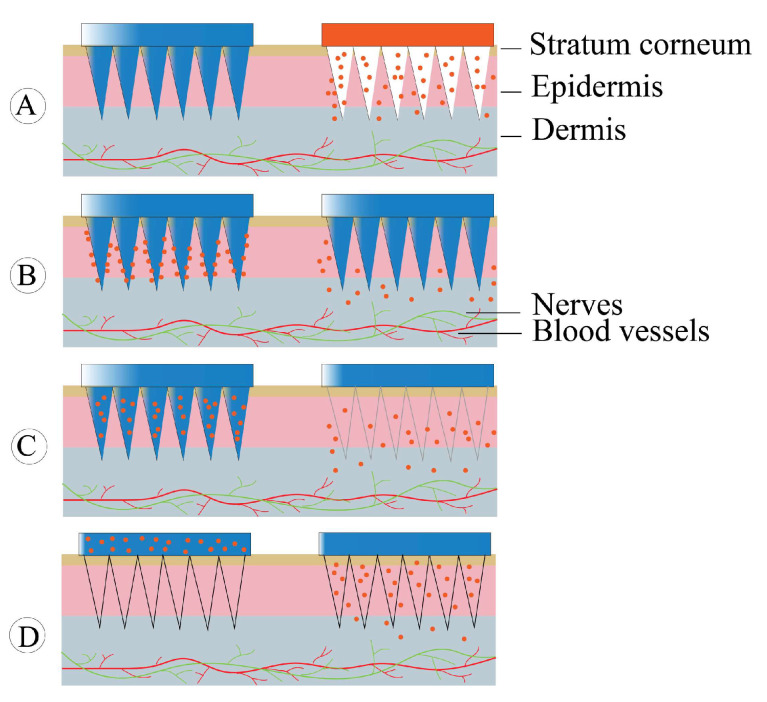
Structural illustration of microneedles available for anti-OP drug delivery: (**A**) solid microneedles; (**B**) coated microneedles; (**C**) dissolving microneedles; and (**D**) hollow microneedles.

**Table 1 molecules-28-06652-t001:** Anti-OP drugs used in the clinic.

Classification	Drug	Dosage Form	Mechanism of Action
Bisphosphonates	Alendronate	Tablets	Specifically binds to hydroxyapatite in bone and inhibits osteoclast activity, inhibiting bone resorption.
	Risedronate	Tablets	
	Zoledronate	Injections	
Hormone	Calcitonin	Injections/nasal sprays	Binds to calcitonin receptors on osteoclasts and inhibits osteoclast activity to reduce serum calcium levels.
Selective estrogen receptor modulators	Raloxifene	Tablets	Activates or inhibits estrogen receptor-mediated cytokine responses, inhibiting bone resorption.
	Bazedoxifene	Tablets	
RANKL inhibitors	Denosumab	Injections	Interfering with the binding of RANKL to its receptor RANK, specifically inhibiting the differentiation and maturation of osteoclasts.
	Strontium ranelate	Dry suspensions	Strontium ions can enhance the DNA synthesis of pre-osteoblasts and promote the multiplication of osteoblasts, increases the expression of osteoprotegerin in osteoblasts, and inhibits bone resorption.
Parathyroid hormone analogs	Teriparatide	Injections	Selectively activate the parathyroid hormone type 1 receptor’s signaling pathway and stimulate osteoblast-mediated bone formation.
	Abaloparatide	Injections	
Sclerostin inhibitors	Romosozumab	Injections	Inhibit the activity of osteosclerosin, promote bone formation, and inhibit bone resorption.
Nutritional supplements	Calcium	Tablets/capsules	The first messenger in the signal transduction pathway, indirectly promoting bone formation.
	Vitamin D		
	Calcitriol		

**Table 2 molecules-28-06652-t002:** Advantages and limitations of different drug delivery strategies.

Strategies	Advantages	Limitations	Ref.
Hydroxyapatite nanoparticles	Biocompatibility, stable mechanical properties, bone affinity.	Poor biodegradability.	[82]
Liposomes	Solubilization, biocompatibility, low immunogenicity, biodegradable, drug protection against external environment degradation, side effects reduction.	Low encapsulation rate, difficulty in scaling up production, high production costs, susceptibility to recognition by the mononuclear phagocytosis system.	[83]
Emulsions	Solubilization, reduction of adverse effects, drugs protection against degradation, better bioavailability.	Low stability, use of surfactants that can be cytotoxic.	[84,85]
Dendrimers	High drug loading capacity, easy surface modification, biodegradable.	Cytotoxicity.	[86]
Micelles	Solubilization, low toxicity, simple preparation.	Sensitive to environmental changes, low stability.	[87]
Polymeric nanoparticles	Easy to chemically modify, drug protection against gastrointestinal environment.	Low physical and chemical stability.	[88]
Scaffolds	Localized drug delivery.	High cost, sudden release, risk of infection.	[79,80]
Microneedles	Fast onset of action, good patient compliance.	Skin irritation, skin sensitization.	[89]

**Table 3 molecules-28-06652-t003:** Nanoparticle systems for anti-OP drug delivery.

Vehicle	API	Drug Loading Method	Outcomes	Ref.
HA-NPs	HA	HA was made into nanoparticles with chitosan or silver.	Reducing serum bone alkaline phosphatase and salivary protein levels.	[91]
HA-NPs	HA	Composite nanoparticles composed of HA-NPs and iron oxides.	Improving the viability of osteoblasts, promoting the expression of Runx2, and inhibiting the activity of osteoclasts.	[93]
HA-NPs	Simvastatin	Simvastatin was loaded in poly(*N*-isopropylacrylamide)-modified mesoporous HA-NPs.	Osteogenic differentiation of BMSCs was promoted and bone formation in OVX rats was improved.	[94]
HA-NPs	Calcitonin	Calcitonin was loaded on HA-NPs by ion complexation.	The relative bioavailability of calcitonin-HA-NPs administered in sublingual mucosa was 15% compared with subcutaneous injection.	[96]
Liposomes	Zoledronic acid	HA was modified on the surface of liposomes, and the drug was encapsulated in a hydrophilic core.	Prolonged release of the drug.	[100]
Liposomes	Icariin	PPi-TEG-Chol is modified on the surface of liposomes to increase their targeting ability.	The bone strength of OVX rats was improved and bone resorption was inhibited to a certain extent.	[104]
Liposomes	Calcitonin	Thioglycolic acid and 6,6′-dithionicotinamide modified chitosan adorn the liposome surface.	The oral bioavailability of calcitonin was 8.2-fold higher than that of free calcitonin solutions.	[106]
Liposomes	Anti-mir-132	The drug was encapsulated in (AspSer)_6_ peptide-modified cationic liposomes.	Liposomes successfully targeted the bone and silenced the expression of miRNA-132-3p in BMSCs, thereby reversing OP.	[108]
Emulsions	Tocotrienols	Self-emulsifying drug delivery system consisting of Cremophor^®^ EL, Labrasol^®^, Captex^®^ 355 and corn oil.	The plasma levels of δ-tocopherols and antioxidant enzymes in the test group were significantly higher than those in the free drug group, which improved the cortical bone thickness and bone strength of OVX rats.	[110]
Emulsions	Raloxifene	Raloxifene nanoemulsions were loaded in a hydrogel composed of poloxamer 407 and carbomer 934 to prolong the adhesion time of milk droplets on the nasal mucosa.	The bioavailability of raloxifene in latex was 7.4 times higher than that of commercially available tablets, and rabbit bone density in the latex group increased by 162% compared to those given oral tablets.	[112]
Emulsions	Teriparatide	Microemulsions consisting of Labrasol^®^, Crodamol GTCC, Solutol^®^ HS 15, D-α-tocopheryl acetate, and aqueous phase (85:15, oil: water).	Bioavailability was 5.4% with oral administration and 12.0% when administered by ileal injection.	[113]
Dendrimers	—	Four amino acids rich in carboxylic acids were coupled to PAMAMs, respectively.	The amount of amino acid-modified PAMAM deposition in the bone was higher than that of unmodified PAMAM.	[47]
Dendrimers	Vitamin D	PAMAM was modified using CH6 aptamers and C11 peptides.	Vehicles were successfully targeted and accumulated in the bone within 24 h after administration.	[117]
Micelles	miR214 antagonist	Asp_8_ peptide was modified on the surface of polyurethane nanomicelles.	After administration, bone mass in OVX mice increased significantly, and osteoclast-associated genes (TRAP and CTSK) were downregulated.	[121]
Micelles	Atorvastatin	Use of tetracycline molecules to modify the surface of PEG-PLGA micelles.	Drug was continuously released from the micelles for more than 48 h, and the bone strength of the TC-PEG-PLGA micelle group significantly increased compared to that of the control group.	[124]
Micelles	Simvastatin	Use tetracycline molecules to modify the surface of PEG-PLGA micelles.	Micelles prolonged the systemic circulation time of simvastatin and preferentially cumulated in the bone tissue	[125]
Polymeric NPs	Risedronate sodium	PLGA nanoparticles were used as carriers.	The accumulative permeability of the drug in the nasal mucosa of pigs was 34.32 ± 2.64%.	[126]
Polymeric NPs	Alendronate sodium	PLGA nanoparticles with surface-modified chitosan and sodium alendronate were used as carriers.	The nanoparticles could continuously release sodium alendronate without obvious synaptic phenomena and had good biocompatibility with MC3T3-E1 cells.	[127]
Polymeric NPs	Calcitonin	Calcitonin and pueraria were encapsulated in chitosan nanoparticles.	The absolute oral bioavailability of calcitonin was up to 12.52 ± 1.83%, which was higher than that of the control group.	[67]
Polymeric NPs	Risedronate sodium	Risedronate was encapsulated in deacetylated chitosan nanoparticles.	The bone density of the rats in the treatment group was significantly improved, the microstructure of trabecular bone was significantly improved, and the cortical bone porosity was small.	[128]
Polymeric NPs	BMP2	The BMP2 gene was encapsulated in chitosan-PEI nanoparticles.	MC3T3-E1 cells in the experimental group were significantly mineralized, and there was formation of new bone in rats with a significant increase in bone defects after administration.	[131]

**Table 4 molecules-28-06652-t004:** Microneedles designed for the delivery of anti-OP drugs.

Type/Technique	Drug	Formulation	Results	Ref.
Solid microneedles /Iontophoresis	Alendronate sodium	Glycerin, itaconate monobutyl ester, 3-sulfopropyl acrylate.	Increased the permeability of the drug.	[152]
Coated microneedles	Calcitonin	Low viscosity CMC-Na, trehalose, poloxamer 188.	There was no significant difference in bioavailability compared to subcutaneous injection, and it was 13 times that of nasal sprays.	[153]
Coated microneedles	Teriparatide	CMC-Na/sucrose.	Sucrose allows the coating layer of teriparatide to dissolve quickly, while CMC-Na slows the drug release rate.	[154]
Coated microneedles	Teriparatide	Titanium, sucrose, hydrochloric acid, EDTA, polysorbate.	The phase II clinical trial found that the microneedle formulation can increase the bone density of the lumbar spine.	[155]
Coated microneedles	Abaloparatide	Zinc chloride	Phase II clinical trial showed a dose-dependent increase in bone density in the spine and hip, but the bone density was lower than with subcutaneous injection.	[156]
Dissolving microneedles	Alendronate sodium	Hyaluronic acid	The decrease in growth plate width and bone density were inhibited.	[157]
Dissolving microneedles	Risedronate sodium, ursolic acid	Gelatin	More than 80% drug release within 24 h in vitro permeation study	[158]
Dissolving microneedles	Teriparatide	Hyaluronic acid	It effectively prevented the reduction of bone density.	[159]
Dissolving microneedles /Iontophoresis	Calcitonin	Maltose	Increasing the blood concentration of the drug.	[160]
Dissolving microneedles	Calcitonin	Silk fibroin, hyaluronic acid.	Trabecular bone repair was better in the preparation group.	[161]
Hollow microneedles	Denosumab	3D-printing technology.	It simulates the release profile of the subcutaneous injection group.	[162]

**Table 5 molecules-28-06652-t005:** Anti-OP preparations in clinical trials. (Data: https://beta.clinicaltrials.gov/ (access on 6 August 2023)).

Drug	Delivery Strategy	Identifier	Phase	Status
Calcitonin	Citric acid acts as a pH adjuster	NCT00959764	Phase III	Completed
Calcitonin	8-[(5-chloro-2-hydroxybenzoyl) amino] octanoic acid (5-CNAC) as an absorption enhancer	NCT00525798	Phase III	Completed
Teriparatide	Transdermal delivery via coated microneedles	NCT01011556	Phase II	Completed
Teriparatide	-	NCT04003467	Phase II	Completed
Teriparatide	Mucosal delivery via nasal sprays	NCT00624481	Phase II	Withdrawn
Teriparatide	Long-acting preparations	CTR20181346	Phase II	Active
rhPTH(1-31)NH_2_	Citric acid acts as a pH adjuster	NCT01321723	Phase II	Completed
Abaloparatide	Solid microstructured transdermal system	NCT04064411	Phase III	Completed
Risedronate sodium	-	NCT02063854	Phase III	Completed

## Data Availability

Data sharing is not applicable to this article.

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
