# Peer review of "Drug Delivery and Therapy Strategies for Osteoporosis Intervention"

_molecules, 2023, doi:10.3390/molecules28186652_

Round 1
Reviewer 1 Report
1. Introduction part at the start can be improved with simple introduction various drug delivery and therapy strategies of OP
2. In targeted drug delivery, brief about the targets of Anti OP drugs in detail
3. In engineering of anti-OP drug write the comparative advantages and disadvantages of fabrication techniques
4. In drug delivery strategies also you can add one table of comparative efficiency of various strategies
5. In microneedle technology multiple types are available so, it is important brief the applications of particular microneedle technology in OP for that you can refer and cite below suggested reference
https://doi.org/10.3390/pharmaceutics14051097
https://doi.org/10.3390/scipharm91020027
6. There are some minor grammatical errors which need to be corrected
7. Improve the resolution of figures
8. Kindly write the long forms of abbreviations used for better understanding of readers
Minor English language editing is required
Reviewer 2 Report
This work systemically reviews the pathogenesis of OP, the engineering of anti-OP drug delivery, novel anti-OP treatment strategies, and clinical anti-OP drug preparations, and especially elaborates several promising drug delivery and therapy systems, including HA-NPs, liposomes, emulsions, dendrimers, micelles, bone tissue engineering scaffolds, and microneedles.
The manuscript is well written.
Because susceptibility and underlying causes of osteoporosis differ between males and females, possibly due to hormonal influences. I suggest authors briefly address such issues involving 1) differences in male and female hormone receptors, 2) the mechanisms of hormonal action, whether medications have varying effects between genders, 3) and whether they result in differing levels of side effects. Probably at the end of the second part.
